# Simulation-Based Multi-Criteria Optimization of Parallel Heat Treatment Furnaces at a Casting Manufacturer

**Thomas Sobottka [1,2,\*], Felix Kamhuber [1] and Bernhard Heinzl [3]**

[1] Fraunhofer Austria Research GmbH, 1040 Vienna, Austria; felix.kamhuber@fraunhofer.at
[2] Department of Industrial and Systems Engineering, Institute of Management Science, Vienna University of Technology, 1040 Vienna, Austria
[3] Research Division of Automation Systems, Institute of Computer Engineering, Vienna University of Technology, 1040 Vienna, Austria; bernhard.heinzl@tuwien.ac.at
\* Correspondence: thomas.sobottka@fraunhofer.at; Tel.: +43-676-888-61626

**Abstract:** This paper presents the development and evaluation of a digital method for multi-criteria optimized production planning and control of production equipment in a case-study of an Austrian metal casting manufacturer. Increased energy efficiency is a major requirement for production enterprises, especially for energy intensive production sectors such as casting. Despite the significant energy-efficiency potential through optimized planning and the acknowledged application potential for sophisticated simulation-based methods, digital tools for practical planning applications are still lacking. The authors develop a planning method featuring a hybrid (discrete-continuous) simulation-based multi-criteria optimization (a multi-stage hybrid heuristic and metaheuristic method) for a metal casting manufacturer and apply it to a heat treatment process, that requires order batching and sequencing/scheduling on parallel machines, considering complex restrictions. The results show a ~10% global goal optimization potential, including traditional business goals and energy efficiency, with a ~6% energy optimization. A basic feasibility demonstration of applying the method to synchronize energy demand with fluctuating supply by considering flexible energy prices is conducted. The method is designed to be included in the planning loop of metal casting companies: receiving orders, machine availability, temperature data and (optional) current energy market price-data as input and returning an optimized plan to the production-IT systems for implementation.

**Keywords:** production planning and control; simulation; optimization; genetic algorithm; heuristics; energy efficiency; heat treatment; scheduling; batching; case study

## 1. Introduction

Heat treatment processes represent one of the most energy-intensive manufacturing processes of cast steel production and usually accounts for ~70% of the total process energy consumption [1]. Thus, increased energy efficiency in steel production is a key factor for competitiveness in global markets, due to long-term rising costs of energy. The manufacturing industry accounts for 36% of $CO_2$ emissions and is responsible for 31% of the corresponding primary energy consumption [2] and digital planning methods are a major potential contributor toward increasing energy efficiency for production companies [3]. A higher level of energy-efficiency can be achieved by either thermal process (parameter) improvements [4–6], optimized demand-side management [7], or production planning [8–12] improvements, e.g., efficient production scheduling. The casting manufacturing system comprises of casting, heat treatment including cooling, and subsequent grinding. According to Wang [1], significant variables and uncertainties exist in the operation of heat treatment equipment,

which have to be considered within scheduling, such as gas heating values, processing times, due dates, and setup-times. Since the availability of ready-to-use digital planning methods for optimized energy efficiency are both crucial for industrial implementations of this planning objective and unavailable for most applications, this paper aims at contributing a case study of a successful development and evaluation of an optimized integrated industrial planning including a presentation of the underlying method and its development.

The contribution at hand focuses on the optimization of parallel heat treatment furnace processes using the upstream casting process as input data. The method is based on a digital model of the real-world system, i.e., a simulation, parametrized with measurements and data from the real system and additional real-time data from the enterprise resource planning (ERP) system (e.g., orders, due-dates, routing tables) as well as variable energy prices for electricity (optional, in some scenarios). The digital model is utilized by an optimization method as the evaluation function. Once the optimized plan is obtained, the plan is fed back into the ERP/MES (manufacturing execution system) for execution, thus closing the loop from real-world system to digital system/twin and back. The planning is meant to operate on a regular basis, e.g., updated planning every day with additional updates in case of changed circumstances (e.g., machine breakdowns or unavailable material), as a rolling horizon planning with a planning horizon of ~1 week.

The simulation itself follows a hybrid discrete/continuous modelling approach that combines discrete material flow simulation with continuous modelling of the energy flows. This makes it possible to accurately model transient energetic processes, such as the heating process of furnaces. Especially in the context of cyber-physical systems (CPS), hybrid simulation is well suited as a tool to model both the cyber part (discrete) and the physical part (continuous) and provide an integrated view for analysis. This integrated view enables to incorporate dynamic effects that lie in the interactions between the physical and the cyber domains and thereby uncover additional potentials for dynamic optimizations. The use case simulation builds on previous works, in particular a hybrid simulation engine together with a library of hybrid modelling components that can be instantiated, connected, and configured for different use cases. This library and some of the components have been extended to incorporate additional requirements imposed by the use case. This includes a crane for loading and unloading the heat treatment furnaces as well adapting the furnace temperature control to time-dependent temperature profiles.

This paper is structured as follows: The following Section 2 provides a review of relevant literature. This is followed by an introduction into the case-study and its requirements in Section 3. In Section 4, the base method developed by the authors in previous work is briefly presented. This is followed by the development of the major components of the case-study specific method presented in Section 5 (simulation) and Section 6 (optimization). Section 7 features the case-study evaluation and discussion of experimental results, followed by a conclusion and outlook in Section 8.

## 2. Background and Related Work

### 2.1. Literature on Optimized Planning Approaches

The paper at hand focusses on the heat treatment process in a casting plant. Thus, the optimized planning considers a batch process and order scheduling/sequencing on parallel machines, with several technological restrictions—e.g., not all products are eligible for processing on all machines—and a complex system behavior, i.e., limited crane handling capacity for loading/unloading of heat treatment furnaces and a complex energy system with heat exchange between considerable material masses, the furnaces and the production hall.

There are several approaches for an optimized batching and scheduling of orders, and they are based on heuristics/rules, metaheuristics, or a combination of both. Yu employed a genetic algorithm (GA) to optimize hybrid flow-shop scheduling and considered machine eligibility [13]. Cheng developed an ant colony optimization for batching in a considerably simpler system of a single machine process [14].

Baykasoğlu covered the cases of parallel heat treatment furnaces as a parallel machine scheduling problem (PMSP), features technological constraints (i.e., not all products can be processed on all furnaces), and introduces a greedy randomized adaptive search [15]. This search algorithm combines a rule-based approach with limited stochastic elements and a local search component. This is meant to increase optimization speed but potentially leaves some global optimization potential through a stochastic search component untapped. Lin develops an iterated hybrid metaheuristic based on the population-based metaheuristic called electromagnetism-like algorithm for a parallel machine scheduling problem [16]. Yilmaz addresses the PMSP with a GA, featuring random key chromosomes, enabling a simultaneously optimized machine assignment and scheduling of orders [17]. Jiang's optimization for the entire steelmaking-continuous casting process chain in a steel production plant is based on differential evolution in a multi-stage optimization with dispatching rules to handle the problem complexity [12]. The heat treatment process and the associated batching is not considered in detail. The approaches listed above solve the order batching and/or the scheduling/sequencing, usually for standardized datasets to compare algorithm performance. They feature simplified system behavior and do not include simulation to consider the dynamic behavior of resources and the energy system.

Approaches featuring simulation can be found in Radim's heuristic and simulation-based scheduling and batching method for parallel heat treatment furnaces [18]. The rule-based batching follows Goldratt's theory of constraints, is applied to a real-world production facility but it does not make use of stochastic optimization methods such as metaheuristics. All approaches mentioned thus far feature objective functions for the optimization with one or two part-goals and do not explicitly consider energy consumption.

Approaches considering energy consumption are the following: Huang presents an energy saving scheduling for a flexible flow-shop production, based on a GA [19], without simulation and without batching. Cheng used the GA for a two criteria batch optimization, pursuing CO2 emission reduction [14]. Wang considered an objective function containing tardiness of orders and energy (gas) consumption, using a GA for batch scheduling of orders in a heat-treatment application [1]. In these approaches, the energy consumption was considered via functions and simplified models without the simulation of complex system behavior.

The following approaches consider the system behavior of the production system (material flow and production process) and the associated energy behavior (e.g., energy consumption and emission of heat) through simulation: Thiede used co-simulation to model the material flow, production equipment, and components of the production and its energy system [20]. The optimization is conducted manually and scenario-based without an automatic optimization module. Rager's approach utilizes a discrete event simulation (DES) in combination with a GA to model the material flow, while the energy consumption is considered with deterministic values [21]. The simulation-based optimization features sequencing and scheduling of orders and does not include the control of equipment; e.g., switching machines on and off. The authors of the publication at hand were co-developers of a simulation-based optimization for a multi-criteria optimization, including energy efficiency, with a comprehensive simultaneous simulation of both energy and production system behavior in a hybrid discrete-continuous simulation method [22]. This approach is the basis for the method developed for the heat treatment use-case, which is introduced in detail in Sections 5 and 6. The base-method is explained in Section 4.

The approaches listed thus far, as representatives of relevant reference approaches in literature, have used heuristics and metaheuristics to solve the NP-hard problem of batching and scheduling of orders on parallel machines. The GA and other evolution-based metaheuristics are the most common successfully applied algorithms. The system behavior is either modelled mathematically within the optimization with simplified features, or simulated, to consider more detailed real-world systems, with the optimization using the simulation as its evaluation function. Concerning the simulation of complex real-life systems there are fewer available approaches and those that are available do not cover batching processes representing an important feature of the heat treatment in the use-case

to be solved. There are examples in the literature for data-based and machine-learning approaches to modelling energy demand [23], however these are currently not suitable for and adapted to the requirements of optimized production planning with the necessary detailed level of system behavior. Nonetheless, the integration of data-based modelling may offer future benefits, e.g., in reducing the computational and modelling effort associated with complex simulations (there have been preliminary steps in this direction for the base method of this paper [24]). Following the literature analysis, a simulation-based optimization with a customized GA was the method chosen for the requirements in this publication—this is also consistent with the result of method comparisons the authors conducted for similar planning problems in the development of the base method presented in Section 4.

*2.2. Literature Concerning Simulation*

There already exists a substantial body of work regarding simulation of production systems for energy efficiency [25–27]. This is driven by an increasing demand from companies for software supporting simulation and integration of energy efficiency performance [28]. Dias [29] provides a comprehensive overview about the most common DES environments. The challenge is to model both the material-flow/production-process behavior and the energy system behavior and their interactions. A simplified way, also found in approaches listed in Section 2.1, is to consider the energy consumption deterministically in a DES. In the use-case at hand, the thermal-physical behavior is important, thus both a discrete and continuous simulation behavior is required. A more comprehensive review of simulation methods and applications used in the manufacturing domain is provided in [30]. The authors conclude an increasing interest in hybrid simulation techniques for handling complex real-world simulation tasks. However, this topic is still only covered by a minority of available publications [26]. Typically, hybrid simulation takes the form of co-simulation [27,28] that couples different simulation environments in a way that the respective sub-models exchange data at given times during a simulation run using a suitable coupling middleware. Co-simulation, however, limits the level of integration between discrete and continuous simulation, the communication across simulation environments is computationally expensive, and the models and their elements are difficult to reuse for other applications since the system boundaries and basic model structure differs in both sub-models [29]. In contrast, our hybrid simulation approach incorporates discrete-event simulation (DES) and continuous simulation models at the building block level of an object-oriented simulation.

The authors of [31] employ a simulation-based approach for system analysis and propose an integrated simulation of interdisciplinary systems, although the models implement a simplified hybridization without differential equations, thereby excluding transient dynamics. In [32], a co-simulation is proposed for a similar application. Other applications for hybrid simulation of continuous and discrete systems can be found in smart grids [33], distributed control systems [34], or healthcare [35].

A different hybrid process modelling approach [6] based on numerical simulation and analytical equations proposes to calculate the radiation, convection, and conduction heat transfer in heat treatment processes to optimize the workpiece loading and the thermal schedule. The furnace and atmosphere temperatures are simplified in this approach to be uniform and equal and disregards the three-dimensional shape of the individual workpieces. The contribution is also focused on the process modelling view only and neglects the production planning and scheduling processes, which is central to the case-study at hand.

## 3. Case Study Introduction

In this section the casting manufacturer case study is introduced as it represents the requirement for the specific planning method to be developed. Figure 1 gives a schematic overview of the main components. The case study is based on an Austrian casting manufacturer and comprises five parallel heat treatment furnaces/ovens of different sizes, capacity, and energy requirements.

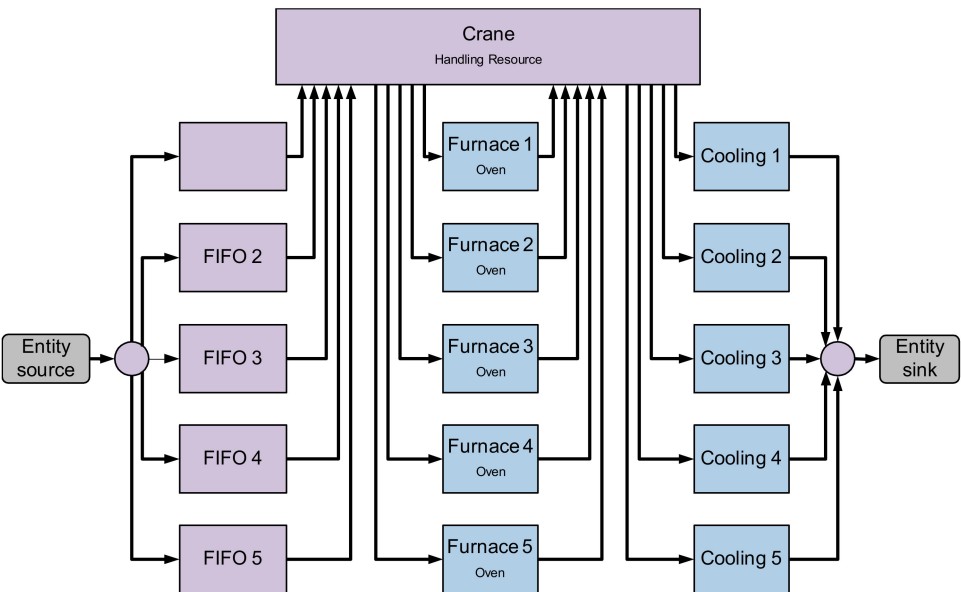

**Figure 1.** Schematic overview of the production layout comprising five heat treatment furnaces with associated cooling stations and first in–first out (FIFO) loading buffers as well as a crane for loading and unloading.

Four of the furnaces are powered by natural gas, while the remaining one uses electric energy for heating. They can perform a total of nine different heat treatment processes (clearing, annealing, warm-up, relaxation, glowing, normalizing, air/water/oil-based steel tempering) differing in their individual temperature profiles and heat treatment durations for each individual article type. These combinations result in roughly 150 different heat treatment programs. Downstream of the furnaces are stations for cooling the workpieces after heating. Depending on the treatment, this takes place in the air or in a water or oil bath. The five furnaces are loaded and unloaded by a single crane that can only operate on one station at a time. Each furnace is loaded from a holding area in front of it that acts as a FIFO (first in–first out) buffer. There, the workpieces are batched and packed onto furnace grates. After treatment, the furnace grates are removed individually to be carried to the subsequent station (cooling). Depending on the temperature requirements of the treatment process, the furnace may still be kept switched on to maintain the temperature. The opening of the furnace door for removing a grate however causes a drop in temperature, which must be compensated again.

The resulting optimization problem is a combination of a batching task and the scheduling and sequencing of orders on parallel machines. The batches, consisting of furnace grates filled with cast iron workpieces, must be created by the planning method in order to be processed. The resulting process consists of the following steps:

1. Combining jobs to batches, which are sequenced and scheduled for processing on the heat treatment furnaces;
2. Transfer of batches to the furnaces via crane;
3. Processing in the furnaces;
4. Release and transfer to cooling stations via crane.

The arrival times of jobs for heat treatment, i.e., the system considered in this paper, are determined by the schedule of the upstream casting process. This casting schedule is provided by the ERP system, which also factors in a considerable amount of rework, since a considerable share of the products have to undergo one or multiple additional heat treatment process iterations, because of quality issues that are only identifiable after the heat treatment process. The actuating variables of the optimization are: order batching, assigning batches to heat treatment furnaces, scheduling/sequencing the processing of

the batches in the furnaces, and controlling the machines (times for switching the machines on and off and thus control the heating process).

## 4. Basic Planning Method

The presented approach is based on the planning method developed in preceding research by the authors [22,36], utilizing hybrid simulation as an evaluation function for an iterative metaheuristic optimization module. The basic optimization procedure and interaction between simulation, optimization, and real-world production planning IT (ERP) are depicted in Figure 2

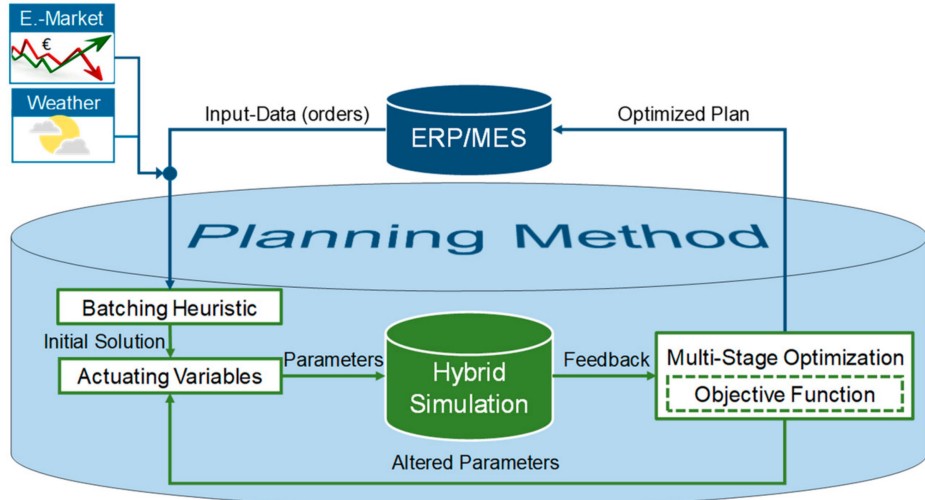

**Figure 2.** Overview of the optimization process.

The ERP/MES provide the main input data for the method: current orders and due dates. This paper is focused on the interaction with ERP/MES systems, since these are the IT systems responsible for production planning and control in a typical industrial automation system landscape [37]. These ERP/MES systems are complemented by SCADA (Supervisory Control and Data Acquisition) software [38] for handling the acquisition and storage of production feedback data. These data are then aggregated and sent to the ERP/MES system, from where they can be used for initialization and alignment of the simulation with the real world.

Together with (optional) variable energy prices (e.g., electricity spot market prices) and weather data (in the current development state this is restricted to outside temperature trend), this is the starting point for the optimization. The optimization initializes simulation evaluations of all intermediate solutions—the simulation feedback in turn is evaluated by the optimization and its objective function, thus informing the next optimization step. This cycle is repeated until a good solution is found and the resulting optimized plan is sent to the planners, ready to be updated into the ERP/MES for execution. The following two subsections summarize the two main components of the planning method: the simulation and the optimization modules. The batching module is part of the case study specific additions to the method that will be presented in Section 6.

### 4.1. Hybrid Simulation

The aim of the simulation is to provide evaluations of different production scenarios in terms of energy usage, production flow, and delivery tardiness. The simulation thus forms the basis for a simulation-supported optimization procedure to optimize production plans for the heat treatment stations while also considering energy efficiency.

In the context of interdisciplinary assessment of energy efficiency in production, it is important to be able to model aspects from different engineering domains with high accuracy. While the material

flow is typically modelled as discrete entities within a discrete-event simulation, energy flow is more accurately described using continuous differential equations. Continuous representation of energy flow, as opposed to discrete energy profiles, enables to accurately incorporate transient dynamics, like for example the heat-up process of a furnace, which have an impact on the overall energy consumption.

To accurately capture the dynamic interactions between these two domains, a hybrid discrete/continuous simulation approach with a tight integration between the discrete and the continuous models is required. Realizing this integration at the component level improves modularity and reusability of hybrid component-based models, which is crucial in practice for managing the complexity of real-world applications [36].

For these reasons, we employ a formal model description, called hyPDEVS [39] for hybrid system modelling [22]. Compared to using typical co-simulation methods, hyPDEVS offers tighter hybrid integration and improved modularity [36] by following a strict component-based paradigm that defines atomic and coupled components, which can be combined to create new application models. This enables modular hierarchical model development of hybrid simulation models that are easily extensible.

From a software perspective, these components can be implemented in an object-oriented manner, in which component classes can be instantiated and reused in different contexts. This enables to implement libraries of model components for various engineering domains. Application engineers then use these pre-defined component models to create application models for new use cases [40].

In the context of the use case in Figure 1, the furnace is an example of such a class (called oven), with different parameter values for each instance to represent the various characteristics (size, capacity, heat transfer, etc.) of the different stations. Another example of a class is the handling resource presented in Section 5.

Based on the hyPDEVS formalism, a standalone hybrid simulator was developed together with a library of components for modelling industrial production systems [22,41]. The components are designed for reusability and take into account entity exchange as well as energy balance equations. More details on the hybrid simulation are described in [39,42].

## 4.2. Metaheuristic Optimization

Production planning optimization problems are NP-hard and thus require approximation methods that provide near optimal solutions in good time. Since the search space in these optimization scenarios typically feature multiple local optima, metaheuristics with their ability to seek global optima with stochastic principles for a wide range of problems have been chosen for the optimization method. Multiple metaheuristics have been compared concerning their optimization performance. A GA, with a set of tuning and customization measures for optimal optimization performance, was chosen as the best fit for the method [43]. The customizations are:

- A guided search by adapted operators in the GA (this prevents the GA from mixing parts of the chromosome that determine different variables and for some variables focusses the mutations in a certain direction, thus reducing the number of practically infeasible solutions);
- A memory function from the Tabu Search algorithm (this prevents the algorithm from evaluating solutions that have already been evaluated before);
- A mixed integer optimization (setting an optimal step size for the optimization significantly reduces the search space without compromising accuracy in practical terms);
- Hybridization by combining the GA with Pattern Search and determining the optimal population size.

The developments and adaptations described in Section 6 adapt the approach to the requirements of the casting manufacturer—mainly comprising new models for complex heat treatment furnaces, optimization strategies to support complex batch-forming processes and an optimization method supporting longer planning horizons in this industry sector.

## 5. Hybrid Simulation for the Case Study

Using the hyPDEVS simulator and component libraries described in Section 4.1, the use case under study (see Figure 1) was implemented in C++. The library includes the model of an industrial furnace for thermal treatment of entities (work pieces, etc.). The model considers the thermal mass of the furnace as well as a product-specific entity mass, including the furnace grate. Given that, the heat transfer of waste heat to the outside as well as heat input into the workpieces are calculated, consisting mainly of heat conduction with a correction factor for radiation and convection.

Compared to other previous use cases, some of which are presented in [41,44], the furnace model had to be extended to incorporate specific requirements: on the one hand, the furnace must be able to follow a given time-dependent temperature profile (instead of just maintaining a constant temperature), which is different for each treatment process. A selection of these temperature profiles is illustrated in Figure 3.

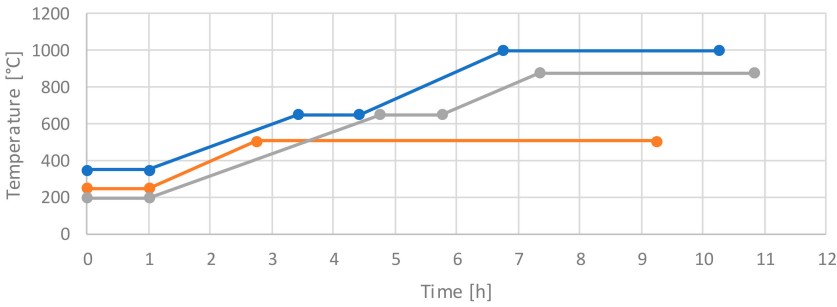

**Figure 3.** Different temperature profiles for different heat treatment processes.

On the other hand, a sudden drop in furnace temperature from opening the furnace door must be considered in the model. This was modelled by a time-dependent heat transfer coefficient, which can be specified over a time series. The furnace station is modelled as a hyPDEVS-coupled component, meaning that it itself is comprised of other, smaller components, representing e.g., the entity queue, thermal cell, or the temperature controller. There are also dedicated (atomic) components for reading and handling the time series values.

The crane for loading and unloading the heating furnaces has been implemented as a new class, see Figure 4. The crane represents a bottleneck in the use case and must consider intervals for loading and unloading as well as during the cooling period. An internal state machine defines the process logic of the class, see Figure 5. From the default state *standby*, an entity – which in this case constitutes a furnace grate—can be picked up at its input *EIN* (state *incoming*), the duration of which is modelled as T1. Subsequently, the crane moves the entity to its destination within a time interval T2 (state *holding*) before it exits at the output *EOUT* (state *output*). When the receiving station has acknowledged the arrival of the entity (signal *EOUTcom*), the crane is blocked for an additional interval T3 (state *waiting*), during which the crane moves back to the starting point.

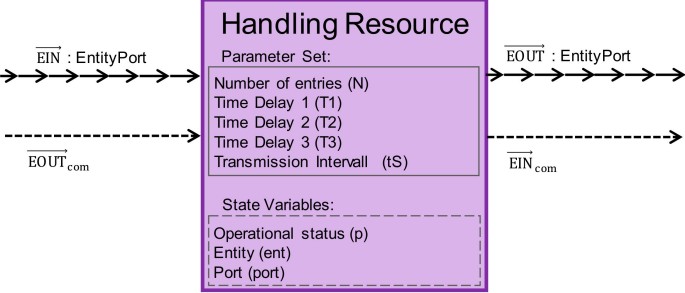

**Figure 4.** Crane component which is implemented as a handling resource class.

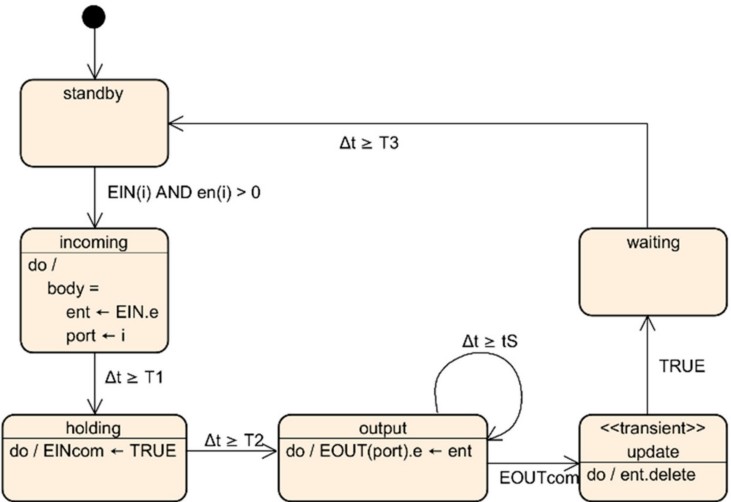

**Figure 5.** State machine for the handling resource.

The state machine is kept simple in order to not unnecessarily complicate the entity flow. It does use a setup duration matrix but assumes uniform durations in the scenarios presented hereafter. It also does not consider any human operator needed to operate the crane and instead assumes ideal availability. Breakdown behavior of equipment can be added later, the main reason to not consider it in this case-study was the lack of data to describe the behavior; a safety margin in the available capacity accounts for production disturbances for now. Furthermore, the model only accounts for the material flow, and is therefore purely discrete. Any energy required for the crane operation is assumed to be negligible and has thus not been modelled.

This abstract modelling on the other hand enables to reuse the handling resource class for other logistical purposes, other than a crane. A handling resource may for example also represent a forklift or an elevator. This makes these components versatile and reusable in different contexts. After implementing the overall use case model, component parameters were calibrated based on production feedback data as well as energy consumption data for a given week in 2018. This ensures that the overall energy demand is depicted correctly, which is relevant for our application, while the modelling simplifications only result in minor deviations in the temporal aspect of the heat transfer.

The model validation was conducted as follows: for each of the five heat treatment furnaces, the simulation model was parametrized with the actual energy consumption (gas and electricity) recorded/measured in multiple reference weeks in 2018, considering the production program executed in those weeks. The parameter values (in particular for the thermal mass $M_{th}$ and heat transmission coefficient UA) were selected in such a way that, for each furnace $i$, the Root Mean Square (RMS) value (1) of the error between calculated and measured consumption, is minimized for all production jobs $j$.

$$RMS_i = \sqrt{\frac{1}{n_i} \sum_{j \in Job\ s_i} \left(W_{j,means} - W_{j,calc}\right)^2}, \tag{1}$$

The resulting parametrization equals the machine-specific physical characteristics, which was implemented in the simulation models for the production equipment (i.e., differential equation systems). The measured/recorded energy consumption values were obtained from the installed data acquisition system in the company and, for gas consumption, were additionally checked against raw meter readings recorded in a different system. Following that, the actual production program that was produced in the reference weeks of our optimization scenario and which is different from the programs in the parametrization reference weeks, was simulated (the actual and not the optimized program was simulated). The simulation results—mainly energy consumption and basic discrete behavior—were then compared with the actual recorded/measured data for the validation period. The results, presented

in Table 1 show that the RMS errors during the validation period are comparable to those during the original parametrization.

**Table 1.** RMS and relative error for parametrization and validation.

|        | UA [W/K] | $M_{th}$ [J/K]       | $RMS_{par}$ | $RMS_{valid}$ | $erel_{valid}$ |
|--------|----------|---------------------|-------------|---------------|----------------|
| Oven 1 | 284.27   | $1.643 \times 10^7$ | 32.95       | 44.87         | −14.16%        |
| Oven 2 | 520.37   | $1.923 \times 10^7$ | 36.76       | 55.05         | −5.14%         |
| Oven 3 | 45.06    | $2.487 \times 10^6$ | 45.81       | 80.79         | −7.06%         |
| Oven 4 | 385.21   | $1.015 \times 10^7$ | 18.07       | 72.91         | 5.72%          |
| Oven 5 | 464.00   | $3.075 \times 10^7$ | 65.78       | 70.81         | −9.72%         |

For most major equipment, we have found deviations of simulated vs. actual/measured energy consumption of below 10%, and no relevant faults in the discrete system behavior. We followed up several of the slightly higher than average deviations (e.g., heat treatment runs for certain product batches and associated heat treatment programs) and could explain the deviations largely with spontaneous deviations from the standard process, e.g., running a heat treatment process for longer than the planned/technologically required duration. The planning and simulation are not meant to simulate these deviations since they are not the approved process and are meant to be minimized in the future. Thus, the validation of the simulation behavior could be confirmed, and the model is ready for use in the optimization. Existing and new heat treatment programs can be reliably simulated, based on their parameters (mainly temperature level and duration) with sufficient accuracy and in flexible scenarios, e.g., with different outside temperatures or with different furnace filling degrees and corresponding thermal masses. This increases accuracy for flexible scenarios, compared with fixed and recorded values, and it also makes future maintenance of the system more efficient, since not every new heat treatment program has to be measured individually before it can be simulated.

## 6. Optimization Method for the Case Study

The major use-case specific requirement for the optimization is the batching of orders and assigning the batches to parallel furnaces under technological restrictions. Orders requiring different temperature profiles, and sometimes additional processes such as hardening, have to be batched into furnace grates, which in turn have to be batched to a furnace processing at one of the available furnaces, each with different capabilities, concerning temperature profiles and additional hardening processes. For this task, a batching heuristic module was developed, creating a valid and initial solution and optimized batches from the input data from the enterprise resource planning (ERP) system. Furthermore, the optimization module is extended by a tailor-made customized problem-specific heuristic optimization module forming a multi-stage optimization together with a GA and a weighted sum [45] based model formalization. The resulting modules of the optimization, which are described in the following subsections, following the model formalization, are:

- A rule-based deterministic batching heuristic;
- A deterministic exchange-based heuristic optimization;
- A GA optimization.

### 6.1. Model Formalization

The multi-stage optimization model consists of a knowledge-based enumerative exchange heuristic and a superimposing GA. It pursues three part-goals that aim at optimizing different planning measures to minimize the manufacturing costs; the first part-goal evaluates the additional costs of shortage and storage of heat treatment output (representing orders $i$ aggregated to oven batches) against their corresponding due dates $t_i$ via a cost-function (in €). The second part-goal, being represented by the

total energy consumption (oil, gas) and its corresponding (constant) energy prices, is calculated and evaluated as cost-term with the feedback of the simulation. The third part-goal evaluates the total load factor of the oven batches $c_j$ versus the maximum possible load factor $c_k$, evaluating the opportunity costs (i.e., business not realized through unused furnace space/capacity) as a linear function of unused capacity. The objective Equation (2), applied in the GA, scalarizes the problem by calculating a weighted and scaled fitness value. The scaling is applied using the part-goals of the best solution from the preceding batching heuristic optimization phase.

$$\text{Minimize } f\left(t_i, \text{e}, \text{g}, c_j / c_k\right) =$$
$$\frac{\omega_1}{s_1} \sum_{i=1}^{m} f_1(t_i) + \frac{\omega_2}{s_2} \sum_{j=1}^{n} \left(e_j k_1 + g_j k_2\right) + \frac{\omega_3}{s_3} \left(1 - \sum_{k=1}^{o} \sum_{j=1}^{n} \frac{c_j}{c_k}\right)$$
$$\omega_1 - \omega_3, s_1 - s_3 \dots \text{part} - \text{goal weightings and scaling factors}$$
$$m, n, o \dots \text{total number of orders, oven batches and furnaces}$$
$$f_1 \dots \text{cost} - \text{function evaluating shortage and storage per order}$$
$$e, g, k_1, k_2 \dots \text{energy consumption and price for electricity and gas}$$
$$c_j, c_k \dots \text{batch load factor and maximum possible load factor}$$

(2)

The part-goal function $f_1$ is calculated based on the completion time on order level ($t_i$) versus its corresponding due date. Early completion times are penalized by storage costs through a flatter partial function, while late completion times are rated by tardiness costs, described by a steeper partial function.

The objective function is accompanied by two constraints that have to be considered during the optimization process. The first constraint does not allow batch orders to start ($t\_start_j$) during weekends, when no staff is available. This means oven batches cannot be started after Saturday morning 05:00 a.m. (represented by $t_{const} = 119\ h\ simulation\ time$), reflecting the shift plan of the plant. This condition is repeated every period length $p$ ($p = 168\ h\ simulation\ time$), resulting in the following mathematical formulation:

$$t\_start_j \ (mod\ p) \le t_{const}, \ \forall j \in \{1, \dots, n\}$$

(3)

The second constraint requires that the combination $s_{j,k}$ of each batch $j$ being selected for the processing on a certain oven $k$ always must result in a valid service combination $\left(s_{j,k} = 1\right)$ for this batch, see Equation (4).

$$s_{j,k} = 1, \ \forall j \in \{1, \dots, n\}, \forall k \in \{1, \dots, o\}$$

(4)

Each batch $j$ has a common heat treatment program. Table 2 describes the relationship between heat treatment and furnace resulting in valid ($s = 1$) and invalid combinations. An invalid combination ($s = 0$) cannot be selected as a possible part of a new solution by the GA.

**Table 2.** Heat treatment to furnace service combinations matrix.

| Heat Treatment | 1 | 2 | 3 | 4 | 5 |
|:---:|:---:|:---:|:---:|:---:|:---:|
| 1 | 0 | 0 | 1 | 1 | 0 |
| 2 | 0 | 0 | 1 | 1 | 0 |
| 3 | 0 | 1 | 1 | 0 | 1 |
| 4 | 1 | 1 | 1 | 1 | 0 |
| 5 | 1 | 1 | 1 | 1 | 1 |
| 6 | 1 | 1 | 1 | 1 | 1 |
| 7 | 1 | 1 | 1 | 1 | 1 |
| 8 | 1 | 1 | 1 | 1 | 1 |
| 9 | 1 | 1 | 1 | 1 | 1 |

The dataset used in the following presentation contains two planning weeks from 2018 (46/2018–47/2018). The prices for electricity and gas represent constant market prices (34 €/MWh for natural gas and 90 €/MWh for electricity). While four oven units are operated with gas, one unit is operated with electricity.

### 6.2. Batching Heuristic

The first step of the heuristic, input-data preprocessing, comprises the cleaning of invalid (furnace overcrowding, ... ) or missing (e.g., material master) input data from the raw ERP data set. The accumulation of order lines into batches to reduce conversions and adjustment of furnaces is based on a similar principle as Lenort's TOC approach (see Section 2) for heat treatment furnaces. The heuristic approach for batching was chosen because of the efficiency and speed, compared to metaheuristic/stochastic approaches, especially since it works without the need for computationally expensive simulation evaluations. Another advantage is that the optimization procedure can be formalized into rules quite well, thus reducing the advantage of a random global search that a GA could provide.

First, the heuristic sorts all orders based on their ERP completion times (derived from planned delivery dates) and order additional priority variable based on information from sales. A furnace occupancy matrix is initialized and for each order the necessary number of grates is calculated. The key principle is to find "batch crystallization" points to initialize batches and to subsequently fill the created batches by iterating over the available orders with increasing order due dates. The batching creation process prioritizes orders requiring specific furnaces and fills the batch with orders requiring the same process (same program number) within an adjustable batch time span (orders in this range of completion dates are eligible for batching in that planning run), set to 72 h for most orders in the case study. Thereby the heuristic tries to utilize the most suitable oven (i.e., reserving scarce capacity for orders requiring high oven capabilities), in terms of size, providing earliest possible free capacity for the to be created batch. The heuristic then iteratively fills up the furnaces offering the remaining capacity to batches that can be processed on all furnaces. Finally, the initial solution for the following optimization steps, consisting of all created batches assigned to specific furnaces, including an estimation of the lead time, is evaluated by the hybrid simulation once.

Between each oven batch, an adjustable interval of one hour is kept for (a) considering the material handling delay of the crane and (b) reheating the furnace to another program-specific temperature—both are considered in detail during the simulation evaluations, but considering them already in the optimization keeps the optimization away from practically infeasible solutions. These two conditions are dependent on the lot-sequence in terms of (heat treatment-specific) reheating and the number of grates on each oven batch. The heuristic, requiring the feedback from the hybrid simulation to be executed, pursues the overall objective of minimizing the intervals between each batch on the corresponding furnace. The visualization of the initial solution created by the batching heuristic is shown in Figure 6.

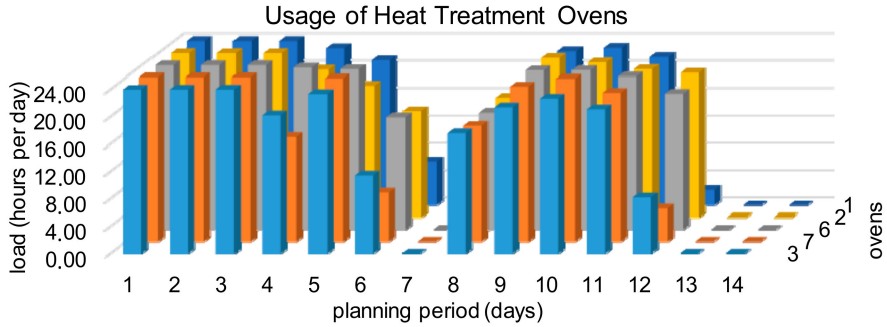

**Figure 6.** Total furnace occupancy.

The results indicate that all five furnaces (also referred to as ovens) are loaded equally in terms of lead-times and total furnace occupancy times. The batch heuristic results are deterministic. In both planning weeks Saturdays feature an uneven load because of the constraint that an oven-batch is not allowed to start after 05:00 a.m. on Saturday.

### 6.3. Hybrid Optimization Approach

A hybrid optimization for the assignment of batches to furnaces and scheduling of furnace processing runs takes over after the batching heuristic, which provides the optimized batches. It consists of a deterministic exchange-based heuristic optimization followed by a customized GA. The exchanged-based heuristic performs a pairwise-exchange of batches between furnaces enumeratively. This procedure is controlled by two parameters: the value of the first parameter defines the maximum amount of time, that exchanged oven batches are allowed to differ in their estimated oven occupancy times (30 min in the scenarios presented herein), while the second parameter is used for the maximum allowed deviation of the estimated starting times between the exchanged batches. The exchange operation is only executed if the respective constraints—oven capacities and heat treatment process restrictions—are met. Furthermore, each solution from the iterative exchange procedure is only kept, if the fitness value, calculated by the feedback of the hybrid simulation, is better than the previous best solution. For the data set at hand and the chosen parameters, this greedy heuristic strategy results in ~900 iterative solution evaluations. The exact value depends on the scope of the respective data set and the chosen values for the two input parameters.

### 6.4. Genetic Algorithm Customization

The GA is implemented in MATLAB® (R2018b) and adapts a GA from the Global Optimization Toolbox. The GA, as the second optimization phase, receives the best solution from the previous deterministic heuristic as the starting point. The implemented stochastic uniform selection operator, ensuring that every individual within a line corresponds to a section of this line length proportional to its scaled value, remains untouched. The algorithm, moving along the line of individuals in steps of equal size, selects the individuals for the next generation stochastically. Because of its stochastic character, this selection operator is suitable for small GA populations. The diversity and exploratory character of the algorithm in this case is preferable to e.g., ranking based or tournament selection operators.

The crossover operator is adapted to create suitable modifications to the sequencing and assignment of the furnace batches. It uses either the pairwise-exchange operator or the single-point-mutation operator to change the time or sequence of an individual. The composition of the batches themselves remain unchanged during the GA optimization.

The parameterization of the GA features, aside from very small population sizes (max. 9 individuals per generation), one elite individual and the disabled utilization of the GA mutation operator. This operator does not add any functionality, as the mutation functionality is already included within the customized crossover operator. The crossover customization features a pairwise-exchange mechanism for oven batches, and an alteration of the start time of batches, thus mimicking the functionality of the GA single-point mutation operator.

Different population sizes are tested within the case study, in combination with a Divide and Conquer (DaC) GA compared with a global search GA: For the DaC GA, a global-search GA with a larger population size and only order/batch swapping allowed, looks for an optimized sequencing, while the second local search GA takes the best individual from the first GA as input, using a smaller population size. The objective of the approach is to first search from a more global perspective to provide an optimized sequence and then, in a second phase, to search for the optimal starting points of each sequenced oven batch. The Divide and Conquer strategy is compared with the results of a global-search GA strategy, in which both sequencing and scheduling are optimized simultaneously (see Section 7.1).

*6.5. Constraint Handling and Validation*

The three model-specific constraints described in the model formalization (see Section 6.1) are implemented as follows: The shift plan constraint (1) and the constraint regarding the furnace service combination matrix (2) are implemented within the optimization routine, while the model-specific crane-unit constraint (3) is covered by the simulation itself, by not allowing the simultaneous handling of more than one batch. The handling-unit specific constraint is modelled in the optimization algorithm by using a minimal distance between each oven batch of half an hour.

The optimization results of test scenarios were validated with the planners of the casting manufacturer. The actual production plans and simulated outcomes were compared with the optimized plan and simulated outcomes. No relevant errors were found, it was confirmed that the optimized plans are practically executable, and that the optimization results are plausible. Together with the already validated simulation behavior, this means that the digital planning method is ready for a proof of concept analysis documented in the next chapter. It is necessary to note that the optimization validation has been conducted with historical data as well as the following experiments. This is a practical limitation for a development in this stage. An actual real-life test for a future planning horizon (i.e., the following week from now) would require implemented live-interfaces between company IT (ERP/MES/control SW), as well as the external data input (e.g., outside temperature forecast and actual temperature as well as optional energy market data). This is not an option mainly due to economic restrictions for all parties involved.

## 7. Experimental Results for the Case Study

The results from the case study include two runs for each scenario of the customized GA. An important part of this research addresses the search behavior of the two GA versions:

- A Global Search GA with different population sizes (small populations with 3, 6 and 9 individuals);
- A Divide and Conquer based, two-phase hierarchical GA, using the optimal population size from the Global Search GA and utilizing only one modification operator in the global and local search phase respectively.

Furthermore, the results cover the influence and potential of variable energy day-ahead price markets. Because of computationally expensive evaluations of the hybrid simulation, only small population sizes are covered by the experimental results section. This is necessary because the algorithm is meant to be used as an operative planning tool, which must provide optimization results within a limited time, e.g., in overnight runs or even shorter update runs in the event of sudden changes (e.g., machine breakdowns or external influences affecting the capacity and material availability). The manual optimization currently implemented in the company is equivalent to the results of the batching heuristic, which is the starting point for the optimization trends for the hybrid heuristic and GA; thus the optimization potential at the end of the simulation runs is roughly equivalent to the potential gains through the planning method over the current manual planning process for the heat treatment.

*7.1. Discussion of the Case Study Results*

The first part of the case study contains the comparison between the GA running with or without the enumerative heuristic. The comparison, illustrated in Figure 7, shows the superiority of the GA in combination with the prior heuristic optimization phase. The heuristic offers the distinct advantage of systematically optimizing the load factor. This results in a 3.5% better performance after 250 evaluation steps when the heuristic and the GA are combined.

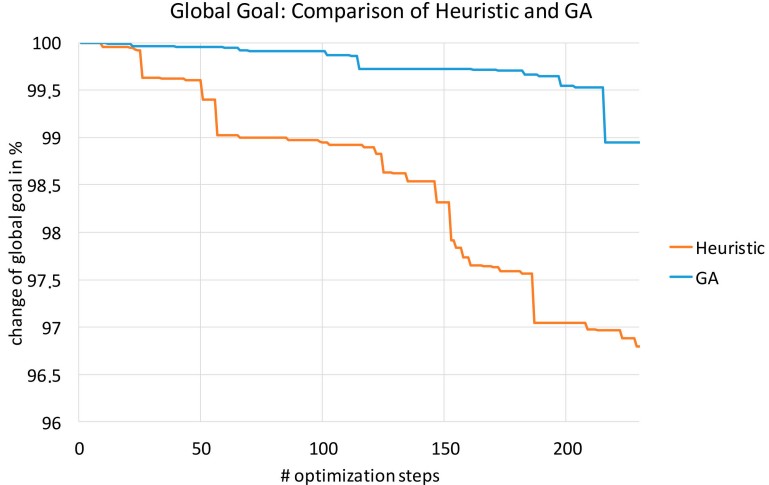

**Figure 7.** Comparison of genetic algorithm (GA) in combination with the heuristic and GA without the heuristic.

The first part of the case study results clearly shows the superiority of very small population-sizes, similar to small population sizes in grey wolf optimizers (GWO). Within the standard GWO the omega wolves of a pack are updated, next to a stochastic random change, in relation to the best three wolves (alpha, beta, and gamma) [46]. The explorative character in the GWO algorithm is expressed by the wolves leaving the original search path to a certain extent and searching into a new direction, while the exploitation is based on the trajectory of the best three wolves. The GA developed herein is based on a very small population (similar to the size of a wolf pack) with the population size varying between 3 and 9 individuals. In the context of long execution times of the hybrid-simulation resulting in a limited number of possible evaluations/optimization steps, the first result (Figure 8) shows the superiority and efficiency of a GA (in runs with 9.000 evaluations each) with a population size of 3 individuals over a GA with a larger population size of 6 or 9 individuals. The global goal improvement of the best parameterization of the GA ensures at least 10% global goal optimization potential.

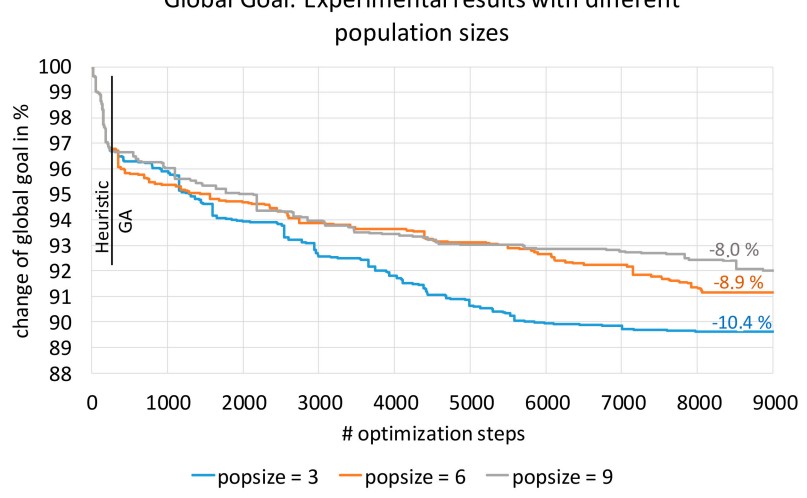

**Figure 8.** Results with different population sizes.

Figure 9 shows the individual part goal trends within the optimization, using the optimized parameterization introduced before. Shortages, carrying a very high penalty, are reduced by 70% within the GA, while the energy costs can be reduced by approximately 6%, and the load factor of the

batches is improved in total by 11%. In almost none of the observed optimization runs can the load factor be further improved by the GA.

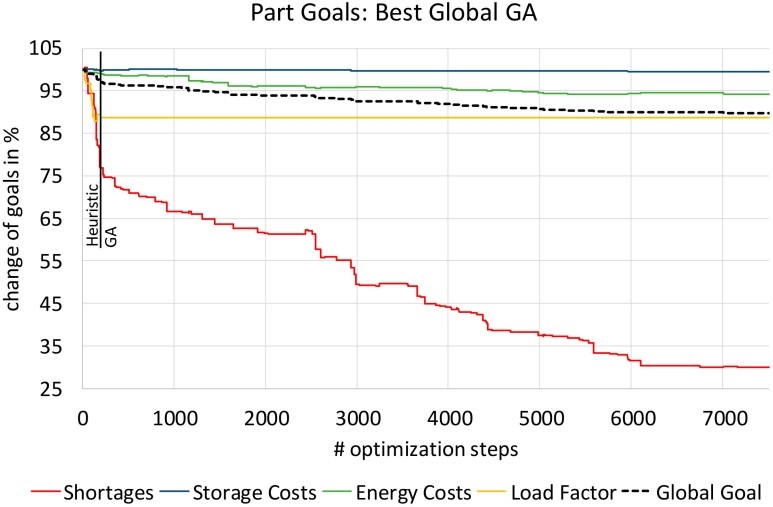

**Figure 9.** Part goal trends.

The total direct $CO_2$ emission potential is derived from the reductions in energy consumption. According to a national electricity-supplier [47] the environmental impact of electricity during the production of the electricity mix is about 170 g/kWh, while natural gas accounts for about 198 g/kWh (combustion calorific value). This value is calculated from the stoichiometric combustion equation for $CH_4$ using the molar mass for $CO_2$. The total direct $CO_2$ emission reduction potential is about 6 tons for the given dataset. This is equivalent to around 200 tons emission reduction per year, depending on the actual furnace utilization.

Figure 10 compares the best parameterization of the global search GA (with population size set to 3) with the sequential hierarchical (Divide and Conquer) GA. The latter utilizes 3 individuals in the first swapping phase and 2 individuals in the following single-point-mutation phase. The figure clearly shows that the sequential GA suffers in its efficiency because of its restricted fitness landscape and opportunities in the corresponding optimization phases.

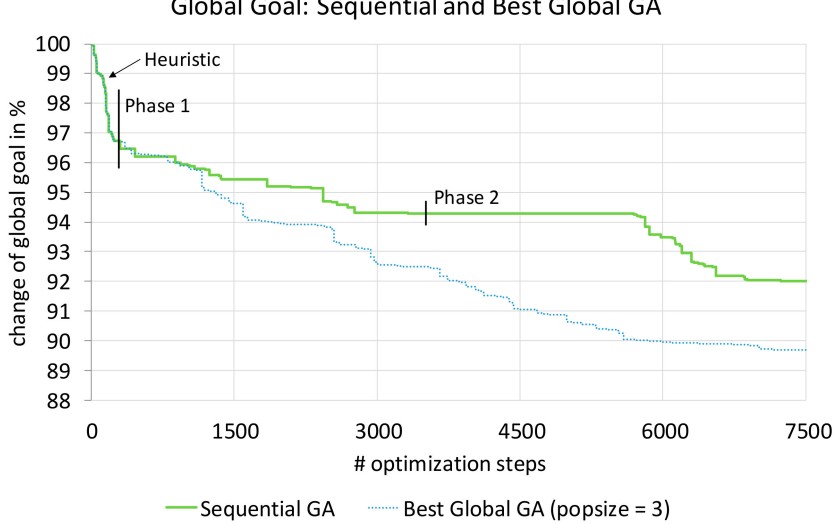

**Figure 10.** Sequential GA and best global GA.

*7.2. Consideration of Dynamic Energy Prices*

In addition to increasing the energy efficiency in the multi-criteria optimization, the planning approach also offers the potential for a better synchronization of energy demand from production sites with the fluctuating energy supply. The energy supply fluctuates significantly over time—especially for electricity and reinforced by the necessary energy transformation toward renewable energy sources, the supply fluctuates significantly every day. If the production sector can better adjust its energy demand through optimized planning, i.e., by considering the availability of energy as input data for an optimized planning, the entire energy system becomes more resource and potentially $CO_2$ efficient. Currently, there are significant hurdles for companies to access the flexible energy trade, most of which are based in the inability for smaller industrial energy consumers ("small" in this case includes industrial production plants as the casting manufacturer in the case study) to predict and plan their demand. This results in fixed energy tariffs for most production plants in Europe—sometimes with a degree of flexibility but never with direct access to long- and short-term energy markets and stock-exchanges. The presented simulation-based approach is both able to better predict and plan the energy demand for the next days and it can consider flexible energy prices as input data for the optimization.

One scenario within the experimental case study covers the consideration of dynamic day-ahead prices [48,49] for electricity and gas, instead of static energy prices. The objective of this scenario is to evaluate the potential of the utilization of a dynamic pricing model versus the current static pricing (flat tariff). Figure 11 shows the price trends for electricity and gas for the time-period of the used dataset. This visualization shows that prices for electricity vary more widely than those for gas and the current fixed tariff for electricity per MWh is above the spot market price for the entire two reference weeks. Table 3 compares the influence of constant and variable energy prices on the optimization outcome. The comparison shows that while there is a cost advantage of using the variable prices, it is relatively small compared to the absolute energy cost and even more so compared to the total cost of all cost categories considered in the objective function. It also shows that for this specific scenario the optimization in its final result (optimized) seems to have opted to sacrifice lower energy costs for the minimization of other cost categories—thus, the relative energy cost advantage of the optimized solution is lower than the advantage in the initial solution. There are several possible explanations: since in this case and for the given datasets the furnaces must run most of the time, the optimization cannot utilize variable prices extensively, e.g., by shifting energy-intensive orders to times of cheaper energy.

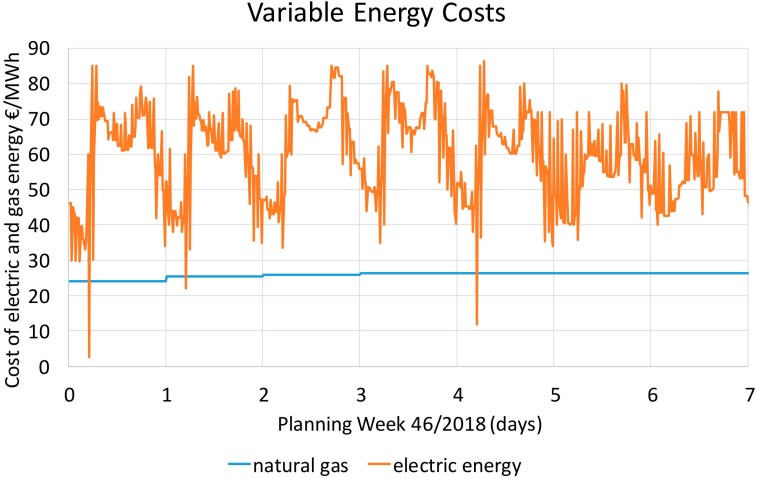

**Figure 11.** Dynamic energy costs (Week 46/2018).

**Table 3.** Influence of variable and constant energy prices (in €).

|  | Variable Price Energy Cost | Contant Price Energy Cost | Advantage of Variable Price for Energy Cost |
|---|---|---|---|
| Initial | 18.185 | 18.411 | 226 |
| **Optimized** | 17.286 | 17.373 | 88 |
| **relative improvement** | 4.94% | 5.64% |  |

The minor variations in natural gas prices do not lead to significant differences in optimization potential. Within this use-case 4 out of 5 heat treatment furnaces are operated with gas. This indicates only limited total optimization potential for flexible energy procurement in this specific case, with the limited considered system-boundaries. For the entire production plant, the total consumption of electricity (two electric arc furnaces for the casting process, machine tools and auxiliary electricity consumers such as hall lighting, etc.) is much higher, which would also increase the potential for advantages due to flexible energy prices. It is also possible to combine multiple industrial energy consumers to larger virtual consumers and there are already service providers that offer this coordination. This clustering of industrial energy consumers could also increase the impact for optimized planning in combination with flexible energy prices.

However, there are additional factors that must be considered when assessing the use of optimized planning with flexible energy prices: there are costs associated with trading on energy markets and engaging in complex energy portfolio management introduces additional risks for the company. A major risk factor is that there can be times of (e.g., externally induced) of high energy prices that can be significantly higher than fixed tariffs and hard to predict. If the planned energy consumption associated with an optimized production planning cannot be fulfilled, e.g., because of internal or external complications, there is a risk of penalties or high surcharges. Also, although tools like the one presented in this paper significantly improve the predictability of energy demand, operating of flexible energy markets would require companies to engage in a complex field requiring expertise in a field usually far removed from the core competences of production enterprises. Thus, the potential for optimized planning in combination with flexible energy management should be invested in more detail to provide actionable advice for companies.

## 8. Conclusions and Outlook

Our contribution shows that the basic concept of a multi-stage hybrid optimization featuring hybrid simulation for the planning task of batching and scheduling, plus assigning orders—batches of orders in this case—to parallel machines is practically feasible and provides considerably better performance than the limited manual planning currently implemented in the company. This is especially important since the automated optimization would be equivalent to hours of manual planning, which is often not practically feasible and had previously not been conducted in comparable detail at the casting manufacturer.

Considering the optimization options, the results show that a combination of an enumerative heuristic and a specific GA is more efficient and robust than a GA implementation only. The global search GA, with a pairwise-exchange operator and the single-point mutation operator combined within the search mechanism, is superior compared to a Divide and Conquer optimization approach. The latter GA approach leaves parts of the search space unexplored in each optimization phase, leading to an inferior solution. The case study at hand shows that GA implementations with small population sizes are more efficient than larger population sizes for the given problem complexity for short optimization runs. This is due to the structure of the large search space, requiring many iterations—since the number of possible simulation-evaluation iterations is limited in practice, smaller population sizes ensure the available iterations are spent on searching for good solutions rather than trying to ensure a more robust optimization behavior.

The hybrid simulation for the energetic evaluation of the production plans does not just rely on fixed energy profiles but performs a physical simulation that considers time-dependent interactions between material and energy flows. For example, loading the furnaces can only be started after pre-heating is completed, which in turn depends on the starting temperature and the environment. Only this integrated approach really does justice to the complexity of modern CPS, in which physical and cyber aspects interact with each other, thereby resulting in emergent behavior.

The method can produce results in time frames suitable for MES/APS and thus can be included in the planning loop of industrial companies. A major challenge for the application is the required quality and detail of data: process information has to be up to date and the initial effort for modelling—especially modelling and parametrization of the physical (thermodynamic) simulation for the production equipment—is demanding for companies, especially given the typical current data maturity level found in the metal casting industry. Future development in the field of digitization and data integration could therefore play a key role in enabling advanced planning tools. The immediate next step would be widening the system boundaries to include the upstream casting process and its large electric arc furnaces, which will increase the optimization potential but also the optimization complexity. Another next step would be validating the method for operative planning, by implementing interfaces for input and output data exchange on a regular basis, which thus far has only been simulated for historical data for a reference period.

The potential of using planning tools like the one presented herein for synchronizing industrial energy demand with fluctuating energy supply has been briefly considered in this case study. A basic feasibility test of the concept has been accomplished and potential savings for the company could be shown. However, this was only a preliminary test and a more in-depth consideration of the practical implementation potential and implications for companies is necessary. This is one of the future directions of this research.

Another future research trajectory is the integration with longer planning horizons, which would enable the method to provide planning support for mid- to long-term production smoothing on the one hand and considering long-term energy markets on the other, completing the support for complex energy portfolio management. Challenges will be data availability and planning with risk, as well as finding suitable modelling simplifications with reduced detail that are derived from the detailed short-term model.

**Author Contributions:** Conceptualization, T.S.; methodology, T.S., F.K., and B.H.; software, F.K. and B.H.; validation, T.S., F.K., and B.H.; formal analysis, T.S., F.K., and B.H.; investigation, T.S., F.K., and B.H.; resources; data curation, T.S., F.K., and B.H.; writing—original draft preparation, T.S., F.K., and B.H.; writing—review and editing, T.S.; visualization, T.S., F.K., and B.H.; supervision, T.S.; project administration, T.S.; funding acquisition, T.S., F.K., and B.H. All authors have read and agreed to the published version of the manuscript.

**Funding:** This research was funded by the Austrian Research Promotion Agency (FFG), grant number 858655.

**Acknowledgments:** The authors would like to thank all case study partners for their contribution, especially the staff at the casting manufacturer in the data acquisition phase.

**Conflicts of Interest:** The authors declare no conflict of interest.

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
