# Peer review of "Simulation-Based Multi-Criteria Optimization of Parallel Heat Treatment Furnaces at a Casting Manufacturer"

_jmmp, doi:10.3390/jmmp4030094_

Round 1

Reviewer 1 Report

The authors presented a planning method of simulation based multi-criteria optimization for a metal casting manufacturer. The paper is well structured and written in general. However, it can be further improved by addressing the following issues. 

First, the objectives of this study are not clearly stated in the abstract and introduction section.  In the first paragraph it says it is contributing a case study while in other places it focuses on the method. The objectives should be consistent throughout the paper. 

Second, in section 6.1, present constrains of variables also in mathematical equations. 

Author Response

We would like to thank the reviewer for the detailed feedback, which we have tried to address in our adaptations. The following points are a summary of our adaptions to your specific comments:

  • We have aligned the goals of the publication in abstract and introduction.
  • The constraints are now presented in mathematical equations.

Reviewer 2 Report

The contribution explains that the basic concept of a multi-stage hybrid optimization featuring hybrid simulation for the planning task of batching and scheduling, plus assigning orders to parallel machines is practically feasible and provides considerably better performance than a manual planning process.

Generally, the topic is interesting and the results are promising.

The problem is the state of the art. There are way too many self-citations and the search for other reserach works is too narrow. The authors need to rework this part and have to include current results of other research groups reported in high-class journals.

Also the description of the interaction with MES and ERP remains superficial without concrete description of the interactions, network, protocols etc..

Author Response

We would like to thank the reviewer for the detailed feedback, which we have tried to address in our adaptations. The following points are a summary of our adaptions to your specific comments:

  • The “live” interfaces MES/ERP were not part of the work and thus we agree that adding specifics is very important for future work concerning the implementation in industry software, which we try to point out validation (6.5) and outlook (8).
  • We have extended the state of the art and included additional surveys and other publications. This brings the number of references to 50, which we perceive to be borderline challenging for a convenient reading-flow for a use-case oriented paper. In our view, width in the literature review leads to including less relevant items and the necessity to explain the relevance – hence, we aim to find an appropriate balance. The two references (both review/general papers) the reviewer has kindly sent us are a good illustration for this:
    The review on EA is an example of a very good but also general, almost textbook-like overview of EAs – this gives readers little additional information for the case at hand, given the fact that specifics concerning EA/GAs are already included in the development of the optimization and in chapter 2.1 there are a number of examples for methods using EA/GA. If this were a publication focused on developing optimization methods, we would (and have in prior work of that kind) included general literature.
    The review on forecasting energy consumption is an attempt at an overview of modelling energy consumption – however, the findings could be mis-interpreted: E.g., data-based approaches according to the review-paper seem to be a common method for factory and multi-machine level considerations. This is not a sensible conclusion for PPC/APS considerations (when product specific sequencing/scheduling decisions have to be made, as well as controlling equipment) – and the publication is not “to blame”, since the goal there was only to “consider energy consumption” on certain system-boundary levels. It tries to select a rather limited number of examples (for a literature review) for a very broad range of possible applications for energy modelling. This leads to the fact that relevant elements (for the use-case at hand) of the scientific discussion, e.g. hybrid/co-simulation and modelling the interactions of energy systems (thermal-physical behavior) and material-flow (production processes) are missing entirely, while the majority of the presented approaches are not very relevant to the specific requirements. We aim at finding the most relevant references for the case at hand.
    Concerning the self-citations: This publication is part of an interdisciplinary research strand across multiple organizations that has been ongoing for several years. The self-citations (few, compared to the number of external references) are an attempt to keep the current paper focused while still enabling readers to check the foundations in prior publications (including a more thorough literature work/review), if desired.

We hope to have found an acceptable compromise between keeping the literature work focused, while still not missing important references for the problem at hand. If there are specific elements that we should include, we remain very grateful for any concrete suggestions.

Reviewer 3 Report

The manuscript covers an interesting topic and fits the scope of the journal. It is well written and organized. Some comments and suggestions are now provided.

The authors put the main focus of data sources in the ERP/MES software. This is correct but it would be improved if SCADA packages were also mentioned due to the fact that data acquisition and storage is commonly performed by these ones and sent to the abovementioned ERP/MES.

The Matlab software is used and the particular toolbox is mentioned. What is the version of these packages? This information could be useful for the interested reader.

Providing both simulation and experimental results is a very good charasteristic of the paper.

The acronym FIFO is used in figure 1 without having been defined.

Some figure captions and titles and tables lack the terminal period (punctuation).

References must be revised tsking into account the template, for instance, the abbreviated title of journals must be used.

Author Response

We would like to thank the reviewer for the detailed feedback, which we have tried to address in our adaptations. The following points are a summary of our adaptions to your specific comments:

  • We have included a mention of SCADA and explain its role in connection with ERP/MES systems.
  • The Matlab Release 2018b and the packages corresponding to this version were used for the modelling and optimization of the industrial use-case.
  • We have added an explanation of the acronym FIFO in the text associated with Figure 1. In addition, we have modified Figure 1 to make it more intuitive to understand and added terminal points for all captions.

Round 2

Reviewer 2 Report

From my point of view, the improvements allow a publication.

Author Response

The authors would like to thank athe reviewers for the productive review process. We are glad that we could adapt the paper satisfactorily.